# Optimal Power Allocation for Channel-Based Physical Layer Authentication in Dual-Hop Wireless Networks

**DOI:** 10.3390/s22051759

**Published:** 2022-02-24

**Authors:** Ningbo Fan, Jiahui Sang, Yulin Heng, Xia Lei, Tao Tao

**Affiliations:** 1China Academy of Space Technology (Xi’an), Xi’an 710000, China; fnb504@163.com; 2National Key Laboratory of Science and Technology on Communication of UESTC, Chengdu 611731, China; 202121220118@std.uestc.edu.cn (J.S.); 17378700918@163.com (Y.H.); 18208107300@163.com (T.T.)

**Keywords:** physical-layer authentication, dual-hop wireless networks, optimal power allocation, false alarm rate, miss detection rate

## Abstract

Channel-based physical-layer authentication, which is capable of detecting spoofing attacks in dual-hop wireless networks with low cost and low complexity, attracted a great deal of attention from researchers. In this paper, we explore the likelihood ratio test (LRT) with cascade channel frequency response, which is optimal according to the Neyman–Pearson theorem. Since it is difficult to derive the theoretical threshold and the probability of detection for LRT, majority voting (MV) algorithm is employed as a trade-off between performance and practicality. We make decisions according to the temporal variations of channel frequency response in independent subcarriers separately, the results of which are used to achieve a hypothesis testing. Then, we analyze the theoretical false alarm rate (FAR) and miss detection rate (MDR) by quantifying the upper bound of their sum. Moreover, we develop the optimal power allocation strategy between the transmitter and the relay by minimizing the derived upper bound with the optimal decision threshold according to the relay-to-receiver channel gain. The proposed power allocation strategy takes advantage of the difference of noise power between the relay and the receiver to jointly adjust the transmit power, so as to improve the authentication performance on condition of fixed total power. Simulation results demonstrate that the proposed power allocation strategy outperforms the equal power allocation in terms of FAR and MDR.

## 1. Introduction

Wireless communication is more vulnerable to eavesdropping and spoofing attacks due to its broadcast nature. Conventionally, the security of wireless networks is addressed by cryptographic protocols above the physical layer that primarily depend on the computation complexity [1]. With the rapid development of advanced computers, wireless networks urgently demand more comprehensive protections that need to be lightweight, flexible, and compatible besides maintaining security [2], especially in the upcoming 6G. Ref. [3] mentioned the trend of using UAV to build cellular networks, and the security of physical layer must be considered.

At present, physical-layer authentication can be divided into three types according to the unique characteristics of extracted signals as follows: (1) authentication based on channel characteristics [4]; (2) authentication based on signal watermarking [5]; (3) authentication based on radio frequency fingerprint [6]. Among them, physical-layer authentication based on channel characteristics is widely studied because of its low computational complexity and broad signal format requirements.

Physical-layer authentication exploits the physical characteristics of channels, devices, and signals to meet the requirements of flexibility and compatibility [4]. The principle of channel-based physical-layer authentication is that the channel response decorrelates rapidly from one transmit–receive path to another if the paths are separated by the order of a wavelength [7]. Specifically, Xiao et al. [8,9] proposed authentication schemes and practical test statistics by analyzing the time and frequency domain of channels. Liu and Wang in [10] proposed an enhanced scheme that integrates multipath delay characteristics into the channel impulse response (CIR)-based physical-layer authentication. With the development of artificial intelligence technology, it was also applied in various fields of communication, including physical-layer authentication. In [11], machine learning was used for physical-layer authentication, and in [12], deep learning was used to optimize UAV trajectory.

In large-scale wireless communication scenarios represented by the Internet of Things and 6G mobile communication network, terminal devices are widely distributed and resource allocation is limited. Power distribution is becoming an issue to be considered. The D2D communication proposed in [13] requires optimal power distribution. End-to-end communication usually requires relays for assistance, and there are only a few research works at present. Zhang et al. in [14] jointly utilized the location-specific features of both amplitude and delay interval of cascaded channels in authentication, while the multipath was assumed to be identical regarding variation in simplifying the consideration; the effects of noise at the relay were not analyzed. We explore the authentication scheme with cascaded channel frequency response based on research on independent subcarriers in the frequency domain, and then we derive theoretical expression of false alarm rate (FAR) and miss detection rate (MDR). Based on the above, we further derive and analyze the way of optimal power distribution.

The remainder of this paper is organized as follows. Section 2 describes the system model. Section 3 describes an authentication scheme with cascaded channel frequency response and a simplified scheme based on majority voting. Section 4 derives the theoretical expressions of FAR and MDR and provides decision threshold under different criteria. Section 5 explores the optimal power allocation by deriving the upper bound for the sum of FAR and MDR. Simulation results and analysis are shown in Section 6. Section 7 concludes the paper.

For the sake of comparison, above schemes are shown in Table 1.

## 2. System Model

As shown in Figure 1, we consider a ubiquitous dual-hop wireless network model with four entities that are represented by Alice (A), Eve (E), Relay (R), and Bob (B). Due to long distance, Alice and Bob can not communicate directly, and node R is required to relay signals. Alice and Eve are in different places, so their signals reach the relay through different buildings, indicating that the two sides pass through different multipath channels in the first hop. Whether the signal is sent by Alice or Eve, Relay amplifies and forwards signals to Bob. Supposing that the last frame Bob received is from Alice, if the new frame Bob receives is from Alice, its channel characteristics will have a strong correlation with the previous frame. Otherwise, if it is from Eve, the channel from Eve to the Relay and the channel from Alice to the Relay are independent, and the channel characteristics will be different from the previous frame so that Bob can use this feature to identify the sender.

Assuming that there are abundant reflectors in the propagation environment, each segment of the cascade channel can be considered as a time-varying multipath channel. Alice sends signal s(t) with Power $P_{1}$, and Relay receives and forwards it to Bob with amplification factor κ. Transmission power of Eve is $P_{1}$, which is supposed to imitate Alice. Thus, the signal Bob receives can be presented as:(1)$$y\left(t\right)=\kappa \sqrt{P_{1}}\sum_{l_{1}=1}^{L_{1}}\sum_{l_{2}=1}^{L_{2}}h_{\ell_{1}}^{X}\left(t\right)*h_{l_{2}}\left(t\right)*s\left(t\right)+\kappa \sum_{l_{2}=1}^{L_{2}}h_{l_{2}}\left(t\right)*n_{1}\left(t\right)+n_{2}\left(t\right)$$
where ∗ is signal convolution, and $h_{l_{1}}^{X}\left(t\right)$ is the channel impulse response of $l_{1}$ multipath of the first hop. $n_{1}$ and $n_{2}$ are additive white Gaussian noises (AWGN) at *R* and *B*, the powers of which are denoted by $N_{1}$ and $N_{2}$, respectively. Relay retransmits the received signal to Bob at power $P_{2}$, and the multiplied amplification factor is:(2)$$\kappa =\sqrt{\frac{P_{2}}{P_{1}+N_{1}}}.$$

The work in [14] obtains multiple independent detection statistics by assuming multipath channels have the same average gain and different delay to improve detection probability. Considering the actual situation, multipath channels usually have different gain levels, and this paper uses the broadband multicarrier transmission mode to obtain multiple channel fading coefficients in the frequency domain to expand the application scenarios. The signal Bob receives in frequency domain can be presented as:(3)$$Y\left(k,t\right)=\kappa H_{RB}\left(k,t\right)\times \left(\sqrt{P_{1}}H_{XR}\left(k,t\right)\times S\left(k,t\right)+N_{R}\left(k,t\right)\right)+N_{B}\left(k,t\right)$$
where $H_{XR}\left(k,t\right)$ and $H_{RB}\left(k,t\right)$ are the channel frequency responses of *X*-*R* and *R*-*B* on the *k*th subcarrier at time *t*.

To describe the temporal variation of channel frequency response in each hop between two adjacent time instants, we employ the auto-regressive model of order 1 in [10], which can be expressed as:(4)$$\left\{\begin{array}{c} H_{XR}\left(k,t\right)=\rho_{1}H_{XR}\left(k,t-1\right)+\sqrt{1-\rho_{1}^{2}}u_{1}\\ H_{RB}\left(k,t\right)=\rho_{2}H_{RB}\left(k,t-1\right)+\sqrt{1-\rho_{2}^{2}}u_{2} \end{array}\right.$$
where $u_{1}$ and $u_{2}$ are complex Gaussian variables that are independent of $H_{XR}\left(k,t-1\right)$ and $H_{RB}\left(k,t-1\right)$ respectively, as denoted by $u_{1}$∼CN(0,1) and $u_{2}$∼CN(0,1). $\rho_{1}$ and $\rho_{2}$ are correlation coefficients between samples spaced by *T* in the first and second hop, given by:(5)$$\left\{\begin{array}{c} \rho_{1}=J_{0}\left(2\pi Tf_{d1}\right)\\ \rho_{2}=J_{0}\left(2\pi Tf_{d2}\right) \end{array}\right.$$
where $J_{0}\left(\;\right)$ is the zero order Bessel function of the first kind, $f_{d1}$ and $f_{d2}$ are the maximum Doppler frequency of two channels, respectively, and T is the time duration of an orthogonal frequency division multiplexing (OFDM) symbol.

Bob receives signal and uses pilot information to estimate frequency response of cascade channel. Without loss of generality, with the least square method, results can be expressed as:(6)$${\hat{H}}_{XB}\left(k,t\right)=\kappa \sqrt{P_{1}}H_{XR}\left(k,t\right)\times H_{RB}\left(k,t\right)+\underset{W}{\underbrace{\kappa {\hat{N}}_{R}\left(k,t\right)\times H_{RB}\left(k,t\right)+{\hat{N}}_{B}\left(k,t\right)}}$$
where ${\hat{N}}_{R}\left(k,t\right)$ and ${\hat{N}}_{B}\left(k,t\right)$ are the estimation error caused by AWGN and can be modeled as complex Gaussian variables with zero mean, and the variance is denoted by $N_{1}$ and $N_{2}$ [9]. In Formula (Equation 6), the first term is the effective term, and the last two terms are equivalent noise terms. ${\hat{H}}_{XB}\left(k,t\right)$ follows complex Gaussian distribution. No matter whether the current message is from *A* or *E*, the second hop it passes through is *R*-*B*, and *B* can extract the channel frequency response of *R*-*B* by exploiting channel estimation technique in [14,15].

## 3. Authentication Scheme Based on Channel Characteristics

### 3.1. Scheme Based on Likelihood Ratio

Supposing that the actual sender of the current signal is *X*, channel frequency response changes $\Delta {\hat{H}}_{XB}\left(k\right)$ such that what Bob obtains from the *k*th carrier can be expressed as:(7)$$\begin{array}{cc} \Delta {\hat{H}}_{XB}\left(k\right) & ={\hat{H}}_{XB}\left(k,t\right)-{\hat{H}}_{AB}\left(k,t-1\right)\\ \; & {=\underbrace{\kappa \sqrt{P_{1}}\left[H_{RB}\left(k,t\right)H_{XR}\left(k,t\right)-H_{RB}\left(k,t-1\right)H_{AR}\left(k,t-1\right)\right]}}_{X_{1}}+\\ & \underset{X_{2}}{\underbrace{\kappa H_{RB}\left(k,t\right){\hat{N}}_{R}\left(k,t\right)-\kappa H_{RB}\left(k,t-1\right){\hat{N}}_{R}\left(k,t-1\right)+{\hat{N}}_{B}\left(k,t\right)-{\hat{N}}_{B}\left(k,t-1\right)}}. \end{array}$$

If the signal is from Alice, $H_{AR}\left(k,t\right)$ and $H_{AR}\left(k,t-1\right)$ are correlated. According to Formulas (Equation 4) and (Equation 7), in case $H_{RB}\left(k,t\right)$ and $H_{RB}\left(k,t-1\right)$ are provided, $\Delta {\hat{H}}_{AB}\left(k\right)$ can be presented as the sum of independent Gaussian variables, which follow complex Gaussian with zero mean, and variance can be presented as:(8)$$\begin{array}{cc} \sigma_{Ak}^{2}= & \frac{P_{1}P_{2}}{P_{1}+N_{1}}{\left(\rho_{1}H_{RB}\left(k,t\right)-H_{RB}\left(k,t-1\right)\right)}^{2}+\frac{P_{1}P_{2}}{P_{1}+N_{1}}\left(1-\rho_{1}^{2}\right)H_{RB}{\left(k,t\right)}^{2}\\ \; & +\frac{P_{2}N_{1}}{P_{1}+N_{1}}\left(H_{RB}{\left(k,t\right)}^{2}+H_{RB}{\left(k,t-1\right)}^{2}\right)+2N_{2}. \end{array}$$

If the signal is from Eve, similarly, the variance of $\Delta {\hat{H}}_{EB}\left(k\right)$ can be presented as:(9)$$\sigma_{E,k}^{2}=P_{2}\left(H_{RB}{\left(k,t\right)}^{2}+H_{RB}{\left(k,t-1\right)}^{2}\right)+2N_{2}.$$

Alice has a correlation at the adjacent time, so $\sigma_{Ak}^{2}$ is smaller than $\sigma_{E,k}^{2}$.

We define $\Delta {\hat{H}}_{XB}\left(k\right)$ as a vector consisting of temporal channel variation from *N* independent subcarriers, which is denoted by:(10)$$\Delta {\hat{H}}_{XB}≜\left[{\hat{\Delta }}_{XB}\left(1\right),\Delta {\hat{H}}_{XB}\left(2\right),\ldots ,\Delta {\hat{H}}_{XB}\left(N\right)\right]$$

Each element of $\Delta {\hat{H}}_{XB}\left(k\right)$ follows Gaussian with zero mean, and its variance is shown in Formulas (Equation 8) and (Equation 9). Each element is statistically independent, so joint probability density function of $\Delta {\hat{H}}_{XB}$ can be presented as:(11)$$f\left(\Delta {\hat{H}}_{XB};H_{n}\right)=\frac{1}{{\left(2\pi \right)}^{N/2}\prod_{k=1}^{N}\sigma_{X,k}}\exp\left\{-\sum_{k=1}^{N}\frac{1}{2\sigma_{X,k}^{2}}{\left({\hat{H}}_{XB}\left(k,t\right)-{\hat{H}}_{AB}\left(k,t-1\right)\right)}^{2}\right\}$$
where n=0,1 are null hypothesis and alternative hypothesis respectively. Then, likelihood ratio functions can be expressed as:(12)$$L\left(\Delta {\hat{H}}_{XB}\right)=\frac{f\left(\Delta {\hat{H}}_{EB};H_{1}\right)}{f\left(\Delta {\hat{H}}_{AB};H_{0}\right)}=\sum_{k=1}^{N}\left(\frac{\sigma_{E,k}^{2}-\sigma_{A,k}^{2}}{\sigma_{A,k}^{2}\sigma_{E,k}^{2}}\right){\left({\hat{H}}_{XB}\left(k,t\right)-{\hat{H}}_{AB}\left(k,t-1\right)\right)}^{2}\overset{H_{1}}{\underset{H_{0}}{≷}}l_{0}$$

Each summation item in Formula (Equation 12) represents change in different subcarriers. Likelihood ratio function in each carrier can be presented as:(13)$$L\left(\Delta {\hat{H}}_{XB}\left(k\right)\right)=\underset{\varphi_{k}}{\underbrace{\left(\frac{\sigma_{E,k}^{2}-\sigma_{A,k}^{2}}{\sigma_{A,k}^{2}\sigma_{E,k}^{2}}\right)}}{\left({\hat{H}}_{XB}\left(k,t\right)-{\hat{H}}_{AB}\left(k,t-1\right)\right)}^{2}$$
where $\varphi_{k}$ is different from each subcarrier, which is related to fading and subcarrier number of two hops. In conclusion, Formula (Equation 13) is sum of squares of independent Gaussian variables that have different variance. It is hard to obtain optimal detection threshold $l_{0}$.

### 3.2. Scheme Based on Majority Voting

To simplify analysis, we make decisions on *N* independent subcarriers respectively. The temporal variation of channel frequency response on the same subcarrier between two adjacent time instants is compared with that of a threshold $Th_{k}$. If it is larger than the threshold, the voting result is recorded as one, which suggests that the current message is more likely from the illegitimate transmitter *E*. On the contrary, if the temporal variation is smaller than the threshold, the result outputs zero. This process can be mathematically expressed as:(14)$$Q_{k}=\left\{\begin{array}{c} 1,\left(\frac{\sigma_{E,k}^{2}-\sigma_{A,k}^{2}}{\sigma_{A,k}^{2}\sigma_{E,k}^{2}}\right){\left({\hat{H}}_{XB}\left(k,t\right)-{\hat{H}}_{AB}\left(k,t-1\right)\right)}^{2}>Th_{k}\\ 0,\;\mathrm{otherwise}\; \end{array}\right.$$
where $Th_{k}$ is the decision threshold. We perform the same operation shown in expression Formula (Equation 14) over *N* independent subcarriers, and then sum up *N* results with equal gain, which are denoted by
(15)$$\Omega =\sum_{k=1}^{N}Q_{k}.$$

Based on the above voting results in Formula (Equation 15), we establish a binary hypothesis testing to differentiate the illegal transmitter *E* from *A*. More specifically, we make the authentication decision by comparing the sum Formula (Equation 15) with a non-negative integer, which can be formulated as: (16)$$\begin{array}{cc} \; & H_{0}:\Omega \leq Z\\ \; & H_{1}:\Omega >Z \end{array}$$
where *Z* is the overall decision threshold.

## 4. Performance Analysis Based on False Alarm Rate and Miss Detection Rate

Considering that FAR and MDR are two fundamental metrics of authentication performance, in this section we analyze FAR and MDR based on [14], and then derive the theoretic decision threshold.

### 4.1. False Alarm Rate

Supposing that the signal is from Alice, Formulas (Equation 7) and (Equation 8) entail that frequency response change $\Delta {\hat{H}}_{AB}\left(k\right)$ follows complex Gaussian with zero mean and variance $\sigma_{A,k}^{2}$. Based on probability statistics, $|\Delta {\hat{H}}_{AB}\left(k\right){|}^{2}$ follows exponential distribution with parameter $\sigma_{A,k}^{2}$, and its probability density function can be expressed as:(17)$$f_{{\left|\Delta {\hat{H}}_{AB}\left(k\right)\right|}^{2}}\left(x\right)=\frac{1}{\sigma_{A,k}^{2}}e^{-\frac{x}{\sigma_{A,k}^{2}}}$$
where $\sigma_{A,k}^{2}$, shown in Formula (Equation 8), is a time-variance function. Based on majority voting, each independent $|\Delta {\hat{H}}_{XB}\left(k\right){|}^{2}$ is multiplied by $\varphi_{k}$, and then compared with threshold $Th_{k}$. So at time *t*, the probability that decision in the *k*th subcarrier outputs one is:
(18)$$\begin{array}{cc} P_{H_{0}}\left(Q_{k}=1|H_{RB}\left(k,t\right),H_{RB}\left(k,t-1\right)\right) & =\int_{Th_{k}/\varphi_{k}}^{\infty }\frac{1}{\sigma_{A,k}^{2}}e^{-\frac{x}{\sigma_{A,k}^{2}}}dx\\ \; & =\exp\left\{-\frac{\sigma_{E,k}^{2}Th_{k}}{\sigma_{E,k}^{2}-\sigma_{A,k}^{2}}\right\} \end{array}$$

Then, the probability that the voting outputs one under null hypothesis is the expectation of Formula (Equation 18) about $H_{RB}\left(k,t\right)$ and $H_{RB}\left(k,t-1\right)$, which is given by:(19)$$P_{H_{0}}\left(Q_{k}=1\right)=E\left\{\exp\left\{-\frac{\sigma_{E,k}^{2}Th_{k}}{\sigma_{E,k}^{2}-\sigma_{A,k}^{2}}\right\}\right\}$$
where E· is expectation of $H_{RB}\left(k,t\right)$ and $H_{RB}\left(k,t-1\right)$. We use $P_{H_{0}}\left(Q_{k}=0\right)$, denoting the probability that the voting outputs zero under null hypothesis, which is the complement of $P_{H_{0}}\left(Q_{k}=1\right)$. Moreover, since noises are white and the transmit power is equally allocated on each subcarrier, the probability $P_{H_{0}}\left(Q_{k}=1\right)$ is identical for all independent subcarriers, and so is $P_{H_{0}}\left(Q_{k}=0\right)$. The probability of *Z* out of *N* outputting one is:(20)$$P_{H_{0}}\left(\Omega =z\right)=\left(\begin{array}{c} N\\ z \end{array}\right)P_{H_{0}}{\left(Q=1\right)}^{z}P_{H_{0}}{\left(Q=0\right)}^{N-z}.$$

So, the theoretical expression of FAR can be expressed as [14]:(21)$$P_{fa}=\sum_{z=Z+1}^{N}\left(\begin{array}{c} N\\ z \end{array}\right)P_{H_{0}}{\left(Q=1\right)}^{z}P_{H_{0}}{\left(Q=0\right)}^{N-z}.$$

### 4.2. Miss Detection Rate

Derivation of the MDR is similar with FAR. We derive probability of each subcarrier and obtain theoretical expression of MDR after majority voting.

Similar with $|\Delta {\hat{H}}_{AB}\left(k\right){|}^{2}$, $|\Delta {\hat{H}}_{EB}\left(k\right){|}^{2}$ follows exponential distribution with parameter $\sigma_{E,k}^{2}$, and its probability density function can be expressed as:(22)$$f_{{\left|\Delta {\hat{H}}_{EB}\left(k\right)\right|}^{2}}\left(x\right)=\frac{1}{\sigma_{E,k}^{2}}e^{-\frac{x}{\sigma_{E,k}^{2}}}.$$

We can obtain the probability that the *k*th voting outputs one (Eve) at time t by integrating the probability density function, which is shown as:(23)$$\begin{array}{cc} P_{H_{1}}\left(Q_{k}=1|H_{RB}\left(k,t\right),H_{RB}\left(k,t-1\right)\right) & =\int_{Th_{♯}\langle \varphi_{k}}^{\infty }\frac{1}{\sigma_{E,k}^{2}}e^{-\frac{x}{\sigma_{E,k}^{2}}}dx\\ \; & =\exp\left\{-\frac{\sigma_{A,k}^{2}Th_{k}}{\sigma_{E,k}^{2}-\sigma_{A,k}^{2}}\right\}. \end{array}$$

Similarly, we can obtain the probability that the voting outputs one under alternative hypothesis, which is given by:(24)$$P_{H_{1}}\left(Q=1\right)=E\left\{\exp\left\{-\frac{\sigma_{A,k}^{2}Th_{k}}{\sigma_{E,k}^{2}-\sigma_{A,k}^{2}}\right\}\right\}.$$

Then, we can derive the theoretical expression based on majority voting [14], which is expressed as:(25)$$P_{md}=\sum_{z=0}^{Z}\left(\begin{array}{c} N\\ z \end{array}\right)P_{H_{1}}{\left(Q=1\right)}^{z}P_{H_{1}}{\left(Q=0\right)}^{N-z}.$$

### 4.3. Decision Threshold

In this section, we analyze two threshold criteria. One is constant FAR threshold based on Neyman-Perarson criterion, whose FAR is obtained by Formula (Equation 21), which can be applied to scenarios with strict requirement on FAR. The other one is a threshold based on minimum error probability criterion. We define *V* as the sum of FAR and MDR:(26)$$V≜P_{fa}+P_{md}.$$

Based on Formulas (Equation 21) and (Equation 25), $P_{fa}$ and $P_{md}$ are the sum of two binomial distributions with different probabilities, so the Formula (Equation 26) is hard to analyze directly. To simplify analysis, we approximate the minimum sum of error rate after majority voting as minimum sum of error rate on single subcarrier. The function can be rewritten as:(27)$$\bar{V}≜P_{H_{0}}\left(Q=1\right)+P_{H_{1}}\left(Q=0\right).$$

Formula (Equation 27) can be regarded as a special case when N=1 and Z=0 in Formula (Equation 26). The channel characteristics of each subcarrier are independent and the distribution characteristics are consistent; thus, changing threshold $Th_{k}$ to let Formula (Equation 27) be the smallest can also make Formula (Equation 26) smallest. Based on Formula (Equation 21) and Formula (Equation 24), the derivative of Formula (Equation 27) with respect to $Th_{k}$ can be rewritten as: (28)$$\frac{\partial \bar{V}}{\partial Th_{k}}=E\left\{-\frac{\sigma_{E,k}^{2}}{\sigma_{E,k}^{2}-\sigma_{A,k}^{2}}\exp\left\{-\frac{\sigma_{E,k}^{2}Th_{k}}{\sigma_{E,k}^{2}-\sigma_{A,k}^{2}}\right\}+\frac{\sigma_{A,k}^{2}}{\sigma_{E,k}^{2}-\sigma_{A,k}^{2}}\exp\left\{-\frac{\sigma_{A,k}^{2}Th_{k}}{\sigma_{E,k}^{2}-\sigma_{A,k}^{2}}\right\}\right\}.$$

Then, we can obtain poles of the Formula (Equation 28):(29)$$Th_{k}=ln\left(\sigma_{E,k}^{2}\right)-ln\left(\sigma_{A,k}^{2}\right).$$

The pole in Formula (Equation 28) is clearly an extremely small point.

## 5. Optimal Power Distribution Scheme Based on Authentication Performance

In dual-hop networks, relay nodes and receivers are often different in both equipment performance and surrounding environment. This section describes how to adjust the power ratio between the sender and trunk nodes to optimize the authentication performance of the system.

### 5.1. Optimized Module

In this section, we can optimize the transmit power allocation between $P_{1}$ and $P_{2}$, improving the authentication performance. More specifically, the optimization problem is to minimize the sum of FAR and MDR on condition of fixed total transmit power, which can be mathematically expressed as:(30)$$\begin{array}{cc} \min_{P_{1},P_{2}} & V≜P_{fa}+P_{md}\\ \;s.t.\; & P_{1}+P_{2}=P \end{array}$$

To simplify the analysis, the objective function is replaced by $P_{H_{0}}\left(Q_{k}=1\right)+P_{H_{1}}\left(Q_{k}=0\right)$, which represents the sum of FAR and MDR when making a decision on a single carrier. The more correct the decision on each subcarrier is, the less that errors will occur in the final combined decision; therefore, the optimization problem is updated to:(31)$$\begin{array}{cc} \min_{P_{1},P_{2}} & V≜P_{H_{0}}\left(Q_{k}=1\right)+P_{H_{1}}\left(Q_{k}=0\right)\\ \;s.t.\; & P_{1}+P_{2}=P \end{array}$$

The optimal threshold provided in Formula (Equation 29) is a time-varying value, and the value for each subcarrier is different, which makes use of the last and present channel estimation of the second hop *R*-*B*. Substituting Formula (Equation 29) into Formula (Equation 31), the objective function can be rewritten as:(32)$$V=E\left\{{\left(\frac{\sigma_{A,k}^{2}}{\sigma_{E,k}^{2}}\right)}^{\frac{\sigma_{A,k}^{2}}{\sigma_{E,k}^{2}-\sigma_{A,k}^{2}}}\left(\frac{\sigma_{A,k}^{2}}{\sigma_{E,k}^{2}}-1\right)\right\}+1.$$

Ignoring the constant term at the end of Formula (Equation 32), the expectation in Formula (Equation 32) is a double integration of Gaussian variables $H_{RB}\left(k,t\right)$ and $H_{RB}\left(k,t-1\right)$, and the item to be integrated is also complicated. We denote the item in the expectation operator as:(33)$$\psi ≜{\left(\frac{\sigma_{A,k}^{2}}{\sigma_{E,k}^{2}}\right)}^{\frac{\sigma_{A,k}^{2}}{\sigma_{E,k}^{2}}}\sigma_{A,k}^{2}\left(\frac{\sigma_{A,k}^{2}}{\sigma_{E,k}^{2}}-1\right).$$

The expression in Formula (Equation 33) can be simplified by taking the logarithm, and a minus sign is added since $\sigma_{A,k}^{2}<\sigma_{E,k}^{2}$, which yields:(34)$$\ln\left(-\psi \right)=\frac{\sigma_{A,k}^{2}}{\sigma_{E,k}^{2}-\sigma_{A,k}^{2}}\ln\left(\frac{\sigma_{A,k}^{2}}{\sigma_{E,k}^{2}}\right)+\ln\left(1-\frac{\sigma_{A,k}^{2}}{\sigma_{E,k}^{2}}\right).$$

After above simplification, the concerned optimization problem can be rewritten as:(35)$$\begin{array}{cc} \; & \min_{P_{1},P_{2}}\frac{N_{1}}{P_{1}}+\frac{N_{2}}{P_{2}}+\frac{N_{1}N_{2}}{P_{1}P_{2}}\\ \; & \;s.t.\;P_{1}+P_{2}=P \end{array}$$

### 5.2. Upper Bound of the Objective Function

In Formula (Equation 35), the double integral of Gaussian variables still exists, which prevents us from finally solving the problem. When solving optimization problems, we usually use the upper bound to replace the objective function that needs to be minimized if the original one is difficult to solve. Here, we provide the proposition $E\left\{\sigma_{A,k}^{2}\right\}/E\left\{\sigma_{E,k}^{2}\right\}$ is the upper bound of the objective function.

**Proof.** We define Δ as the difference between $\sigma_{A,k}^{2}$ and $\sigma_{E,k}^{2}$ to simplify the function, which can be rewritten as:
(36)$$\Delta ≜\sigma_{E,k}^{2}-\sigma_{A,k}^{2}=\frac{2\rho_{1}P_{1}P_{2}}{P_{1}+N_{1}}H_{RB}\left(k,t\right)\cdot H_{RB}^{*}\left(k,t-1\right)$$$H_{RB}\left(k,t-1\right)\approx H_{RB}\left(k,t-1\right)$, $H_{RB}\left(k,t-1\right)$ and $H_{RB}\left(k,t\right)$ are in coherent time. Thus, the first derivative of Δ with respect to $H_{RB}\left(k,t\right)$ can be represented as:
(37)$$\frac{\partial \Delta }{\partial H_{RB}\left(k,t\right)}=\frac{4\rho_{1}P_{1}P_{2}}{P_{1}+N_{1}}H_{RB}\left(k,t\right)$$Based on Formula (Equation 9), the first derivative of $\sigma_{E,k}^{2}$ with respect to $H_{RB}\left(k,t\right)$ can be expressed as:
(38)$$\frac{\partial \sigma_{E,k}^{2}}{\partial H_{RB}\left(k,t\right)}=4P_{2}H_{RB}\left(k,t\right)$$Comparing Formulas (Equation 37) and (Equation 38), we can derive that:
(39)$$0<\frac{\partial \Delta }{\partial H_{RB}\left(k,t\right)}<\frac{\partial \sigma_{E,k}^{2}}{\partial H_{RB}\left(k,t\right)}.$$As $H_{RB}\left(k,t\right)$ goes up, $\sigma_{E,k}^{2}$ goes up, while $\Delta /\sigma_{E,k}^{2}$ goes down. Thus, the correlation between the two is less than zero, which can be rewritten as:
(40)$$\begin{array}{cc} \mathrm{cov}\left(\sigma_{E,k}^{2},\frac{\Delta }{\sigma_{E,k}^{2}}\right) & =E\left\{\sigma_{E,k}^{2}\times \frac{\Delta }{\sigma_{E,k}^{2}}\right\}-E\left\{\sigma_{E,k}^{2}\right\}E\left\{\frac{\Delta }{\sigma_{E,k}^{2}}\right\}\leq 0\\ \; & \Leftrightarrow E\left\{\sigma_{E,k}^{2}\right\}E\left\{\frac{\Delta }{\sigma_{E,k}^{2}}\right\}\geq E\left\{\Delta \right\} \end{array}$$Substituting (Equation 36) into (Equation 40), we can obtain the following inequation:
(41)$$E\left\{\frac{\sigma_{A,k}^{2}}{\sigma_{E,k}^{2}}\right\}\leq \frac{E\{\sigma_{A,k}^{2}\}}{E\{\sigma_{E,k}^{2}\}}$$ □

In Rayleigh channel, correlation coefficient of $H_{RB}\left(k,t\right)$ and $H_{RB}\left(k,t-1\right)$ with zero mean and one variance is $\rho_{2}$ in Formula (Equation 5). Thus, $E\{\sigma_{A,k}^{2}\}$ can be written as: (42)$$\begin{array}{c} E\left\{\sigma_{A,k}^{2}\right\}=\frac{P_{1}P_{2}}{P_{1}+N_{1}}\left[\rho_{1}^{2}E\left\{H_{RB}{\left(k,t\right)}^{2}\right\}-2\rho_{1}E\left\{H_{RB}\left(k,t\right)H_{RB}^{*}\left(k,t-1\right)\right\}+E\left\{H_{RB}{\left(k,t-1\right)}^{2}\right\}\right]\\ +\frac{P_{1}P_{2}}{P_{1}+N_{1}}\left(1-\rho_{1}^{2}\right)E\left\{H_{RB}{\left(k,t\right)}^{2}\right\}+\frac{P_{2}N_{1}}{P_{1}+N_{1}}\left[E\left\{H_{RB}{\left(k,t\right)}^{2}\right\}+E\left\{H_{RB}{\left(k,t-1\right)}^{2}\right\}\right]+2N_{2}\\ =2P_{2}+2N_{2}-\frac{2P_{1}P_{2}\rho_{1}\rho_{2}}{P_{1}+N_{1}} \end{array}$$

Similarly, $E\{\sigma_{E,k}^{2}\}$ can be written as:(43)$$E\left\{\sigma_{E,k}^{2}\right\}=P_{2}\times \left(E\left\{H_{RB}{\left(k,t\right)}^{2}\right\}+E\left\{H_{RB}{\left(k,t-1\right)}^{2}\right\}\right)+2N_{2}=2P_{2}+2N_{2}.$$

Thus, the upper bound of objective function can be represented as:(44)$$\frac{E\left\{\sigma_{A,k}^{2}\right\}}{E\left\{\sigma_{E,k}^{2}\right\}}=\frac{2P_{2}+2N_{2}-\frac{2P_{1}P_{2}\rho_{1}\rho_{2}}{P_{1}+N_{1}}}{2P_{2}+2N_{2}}=1-\frac{\rho_{1}\rho_{2}}{1+\frac{N_{1}}{P_{1}}+\frac{N_{2}}{P_{2}}+\frac{N_{1}N_{2}}{P_{1}P_{2}}}.$$

### 5.3. Approximate Optimal Solution

In (Equation 44), parameters of upper bound $E\left\{\sigma_{A,k}^{2}\right\}/E\left\{\sigma_{E,k}^{2}\right\}$ contain power of the sender $P_{1}$, power of the relay $P_{2}$, noise power of the relay $N_{1}$, noise power of the receiver $N_{2}$, correlation coefficient in first hop $\rho_{1}$, and correlation coefficient in second hop $\rho_{2}$, where $P_{1}$ and $P_{2}$ are adjustable variables. Usually, $\rho_{1}$ and $\rho_{2}$ remain unchanged for a period of time; thus, $\rho_{1}$ and $\rho_{2}$ in Formula (Equation 44) have little influence on power distribution. As a result, we ignore $\rho_{1}$, $\rho_{2}$ and constant item in (Equation 44), so the optimization problem can be written as:(45)$$\begin{array}{cc} \min_{P_{1},P_{2}} & \frac{N_{1}}{P_{1}}+\frac{N_{2}}{P_{2}}+\frac{N_{1}N_{2}}{P_{1}P_{2}}\\ s.t. & P_{1}+P_{2}=P \end{array}$$

We define μ as the ratio of transmission power and total power, and *T* as objective function in (Equation 45). Thus, *T* can be written as:(46)$$T=\frac{\left(N_{2}P-N_{1}P\right)\mu +N_{1}P+N_{1}N_{2}}{\left(\mu -\mu^{2}\right)P^{2}}.$$

We take the derivative of *T* with respect to μ and let it equal to 0:(47)$$\frac{P^{2}\left[\left(N_{2}P-N_{1}P\right)\mu^{2}+\left(2N_{1}P+2N_{1}N_{2}\right)\mu -N_{1}P-N_{1}N_{2}\right]}{{\left[\left(\mu -\mu^{2}\right)P^{2}\right]}^{2}}=0.$$

Combining three cases of $N_{1}$ and $N_{2}$, we can write the power allocation μ as:(48)$$\mu_{0}=\frac{P_{1}}{P}=\frac{\sqrt{N_{1}P+N_{1}N_{2}}}{\sqrt{N_{1}P+N_{1}N_{2}}+\sqrt{N_{2}P+N_{1}N_{2}}}.$$

## 6. Simulation Results

In this section, for the purpose of validating the theoretical results of Section 4 and Section 5, we use MATLAB to simulate the theoretical results.

We define the signal-noise ratio (SNR) of the dual-hop wireless networks in the concerned scenario as the total power transmitted to the noise power, given by:(49)$$\mathrm{SNR}=\frac{P_{1}+P_{2}}{N_{1}+N_{2}}.$$

The key simulation parameter settings are illustrated in Table 2 and Table 3.

As shown in Table 2, carrier frequency, subcarrier interval and channel parameters used in the table are typical LTE system parameters [16]. In addition, the number of subcarriers corresponds to the minimum bandwidth of 1.25 MHz in LTE. In fact, with the increase in bandwidth, the number of independent subcarriers that can be obtained in the frequency domain increases, which will be more favorable to the algorithm in this paper. The false alarm probability of identity authentication is selected as a typical value of 5%.

Without loss of generality, the coherent bandwidth can be calculated by the parameters in the Table 2. Moreover, independent subcarriers can be selected, and the total transmit power is assumed to be 1.

To explore the difference in authentication performance between likelihood ratio test (LRT) and majority voting algorithm (MV), we compare them in terms of the probability of detection while keeping FAR constant as 0.05. The threshold involved in MV is theoretically derived, while the decision threshold in LRT is found by exhaustive method to keep FAR constant. We also attempt to find the threshold while ignoring the influence of cascade channel, as well as the threshold based on single-carrier threshold multiplied by the number of independent subcarriers. The simulation results are shown in Figure 2 (the vertical axis represents the detection probability, and the horizontal axis the signal-to-noise ratio).

As shown in Figure 2, MV is better than two experimental LRTs, while exhaustive is better than MV. Because the temporal channel variation on different subcarriers can be summed up in LRT, which has a smooth effect. While the decision on each subcarrier can be regard as one-bit quantization, and some precision is lost. The gap between exhaustive LRT, experimental-1 LRT, experimental-2 LRT, and MV reduces in the high SNR region, where the detection probability is more than 95%, meeting the requirements of general systems.

To validate the theoretical expression for FAR and MDR, derived in Formulas (Equation 21) and (Equation 25), we compare them with simulation results. In MV algorithm, we can adjust the decision threshold to realize constant FAR as needed, and the probability of detection with different FAR is shown in Figure 3 (the vertical axis represents the detection probability and the horizontal axis the signal-to-noise ratio).

In Figure 3, the theoretical results are consistent with simulation results under different parameters, which prove the correctness of the formulas Formulas (Equation 21) and (Equation 25). The authentication based on majority voting algorithm can be a theoretically analyzed performance, which is a major advantage over LRT and also makes it more practical. Under constant false alarm condition, the missed detection probability tends to a minimum value with the increase in SNR by optimizing the threshold.

To validate minimum error probability threshold proposed in Formula (Equation 29), we compare sum of FAR and MDR in three simulation scenarios that are optimal: 5% FAR and 3% FAR. As shown in Figure 4, optimal threshold is below the other two curves, which meets its physical meaning. In addition, two curves with different FARs intersect, since at low SNR, the difference between legal and illegal transmitter is small, causing MDR to go down as FAR goes up, and this is the opposite case when SNR is high.

To prove the universality of the threshold Formula (Equation 29), comparative analysis was conducted in several different scenarios, which are characterized by the noise power at relay since the total noise power was controlled as one. In Figure 5, authentication performance of theory and exhaustion fits perfectly in three scenarios.

To compare authentication under two power allocation schemes, we perform simulation at different SNRs, and results are shown in Figure 6. As shown in Figure 6, optimal scheme is better than equal allocation, especially at low SNR, since difference between legal and illegal transmitter has more influence than noise at high SNR.

To validate the performance of the theoretically approximate optimal scheme, we compare it with exhaustive optimal scheme in Figure 7, where theoretical approximate optimal scheme is derived in Formula (Equation 48) and exhaustive optimal scheme is designed to exhaust power allocation with small granularity. As shown in Figure 7, the gap between the practical minimum sum of FAR and MDR and the sum caused by the proposed power allocation is about 0.002. The small gap implies that Formula (Equation 48) is approximately optimal and effective in practice.

## 7. Conclusions

This paper explored channel-based physical-layer authentication in dual-hop wireless networks. By analyzing the characteristics of cascaded channel, we established the likelihood ratio test (LRT) at first. To simplify, the majority voting algorithm was employed. Based on this simplification, we derived the theoretical expressions for false alarm rate (FAR) and miss detection rate (MDR), and we analyzed the upper bound for their sum. Moreover, we proposed an optimal decision threshold that utilized the channel estimation of the second hop to provide a more accurate decision. With this threshold, the optimal power allocation minimizing the sum of FAR and MDR was derived. In addition, it is expected that the proposed power allocation is useful and provides a novel mode of thought in optimizing dual-hop physical-layer authentication. When in a mobile state, the authentication performance based on channel characteristics declines. Adjusting the number and position of pilots used for authentication can optimize the performance. In addition, the algorithm can be further optimized by channel state prediction and other technologies.

To sum up, in 6G large-scale heterogeneous network, there are a large number of devices with different upper-layer access protocols.The physical-layer authentication technology is transparent to the upper-layer protocols, and thus it has good compatibility and can complement the existing upper-layer traditional security schemes to jointly build a more comprehensive security system.

## Figures and Tables

**Figure 1 sensors-22-01759-f001:**
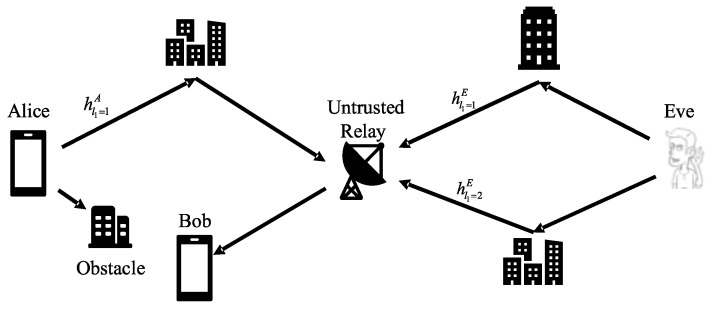
System model: Alice communicates with Bob with aid of amplify and forward (AF) relay; Eve is a would-be intruder impersonating Alice.

**Figure 2 sensors-22-01759-f002:**
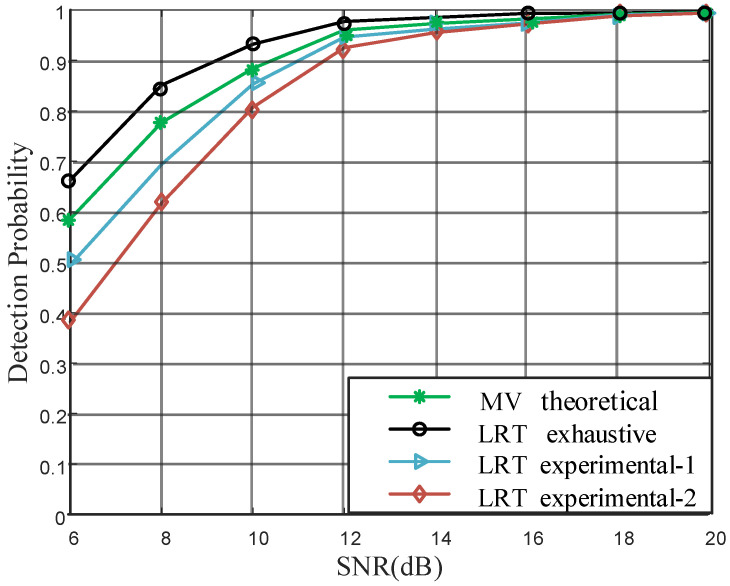
Likelihood ratio test (LRT) and majority voting algorithm comparison.

**Figure 3 sensors-22-01759-f003:**
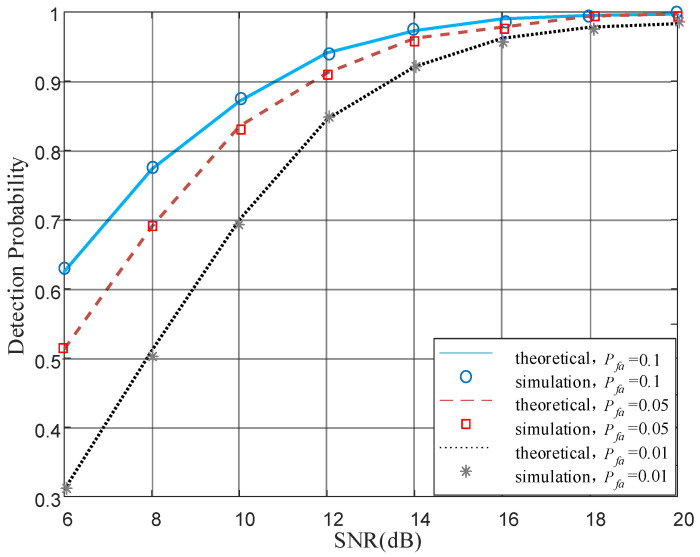
Theoretical expression for false alarm rate (FAR) and miss detection rate (MDR) validation.

**Figure 4 sensors-22-01759-f004:**
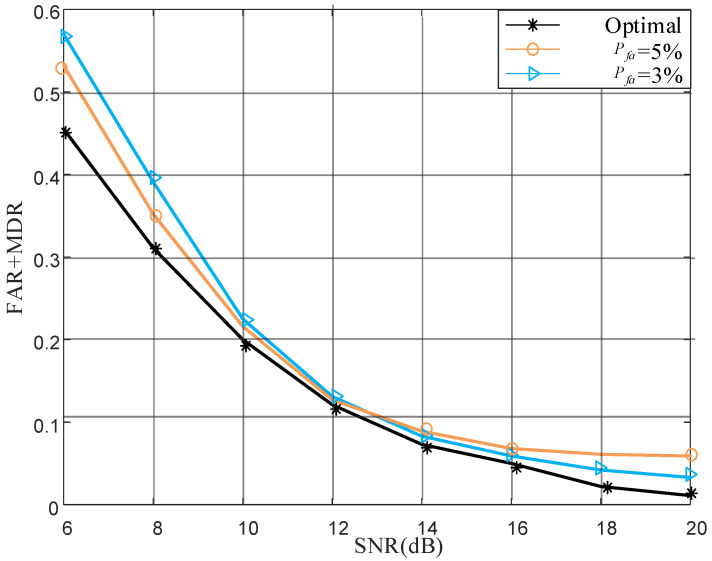
Authentication performance at different thresholds.

**Figure 5 sensors-22-01759-f005:**
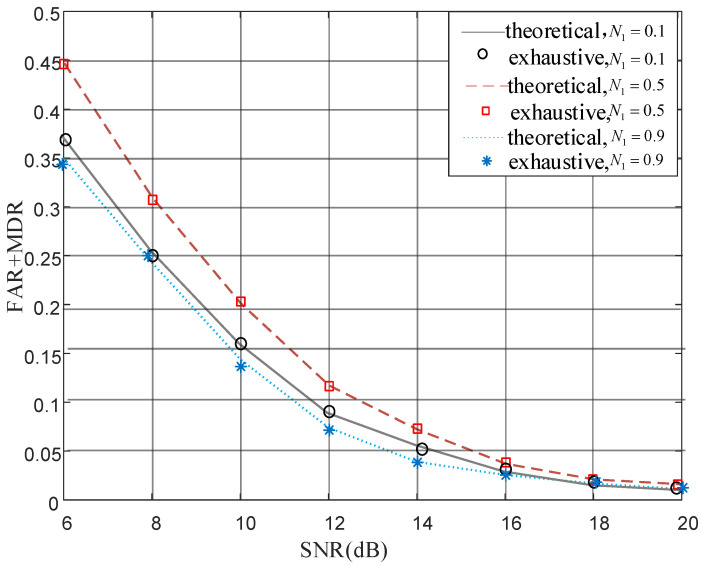
Authentication performance in different relay noise power.

**Figure 6 sensors-22-01759-f006:**
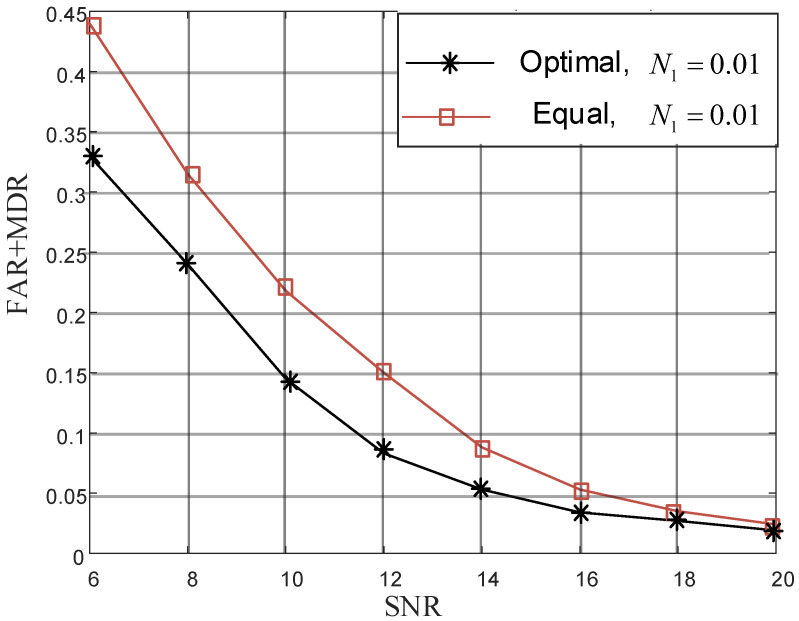
Performance comparison of two allocation schemes.

**Figure 7 sensors-22-01759-f007:**
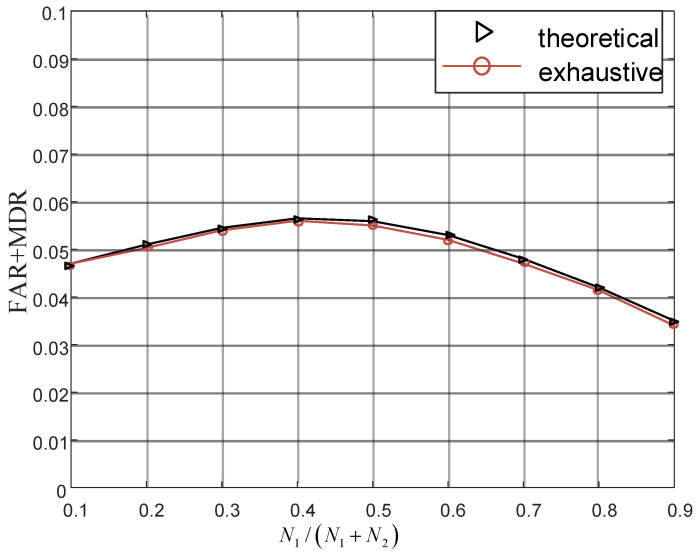
Optimal threshold validation.

**Table 1 sensors-22-01759-t001:** Comparison of several authentication schemes.

	Authentication based on time domain multipath channel characteristics	Authentication based on frequency domain multipath channel characteristics	Authentication based on signal watermarking	Authentication based on fingerprint identification
Principle	Based on time domain impulse response	Based on frequency domain impulse response	Transmit secret security authentication code or label with message	Based on physical differences of analog equipment
Main defects	Limited by the number of multipaths and the ability to distinguish	Limited by Doppler changes	Low power efficiency; watermark has an impact on the main signal	Characteristics are random and weak; poor stability

**Table 2 sensors-22-01759-t002:** System parameters.

Carrier frequency	2.4 GHz
Subcarrier interval	15 kHz
Number of subcarriers	128
Relative speed	200 km/h

**Table 3 sensors-22-01759-t003:** Channel parameters of extended vehicular A (EVA) model.

Number of multipaths	9
Multipath time delay (ns)	(0 30 150 310 370 710 1090 1730 2510)
Multipath relative power (dB)	(0 −1.5 −1.4 −3.6 −0.6 −9.1 −7.0 −12.0 −16.9)

## Data Availability

Not applicable.

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
