# Peer review of "Optimal Power Allocation for Channel-Based Physical Layer Authentication in Dual-Hop Wireless Networks"

_sensors, 2022, doi:10.3390/s22051759_

Round 1

Reviewer 1 Report

In this work, the authors present a physical-layer authentication scheme for dual-hop wireless networks. The proposed scheme utilizes the characteristics of the channel, while two authentication schemes are presented, namely the likelihood ratio detection scheme and the majority voting algorithm. The authors provide an overview of the system model and present in detail the operation of the channel-based authentication mechanism. Moreover, they derive analytical expressions for the two detection schemes and evaluate their performance in terms of false alarm and miss detection rates. 

The work is focused on a timely and critical topic, as dual-hop, relay networks, and device-to-device communications are expected to be extensively leveraged in 5G and beyond mobile networks. The operating principles of the two schemes are thoroughly presented, while analytical expressions are also derived. Moreover, an optimal power allocation algorithm is designed in order to further improve the performance of the authentication mechanism. In this respect, the evaluation results show that the employed schemes feature a high level of performance. Moreover, the structure and organization of the paper, as well as the use of the English language, are considered good. Nevertheless, there are some recommendations to be considered as follows:
1) The overview of the state-of-art (related works) should be expanded with more details and additional research works.
2) For completeness, it would be interesting to describe the impersonation process in the description of the system model (figure 1).
3) Some of the acronyms have not been defined (e.g., AF in figure 1 caption, OFDMA in line 73, etc.). The authors are kindly requested to recheck the definitions of all acronyms.
4) The text below equation (2) should be revised as: 'The work in [14] obtains ...".
5) A table summarizing all notation and symbols used throughout the paper would assist the readers.
6) More details on the link-level simulator can be added. For example, what software was utilized?
7) In figures 2 and 3, what does the y-axis label (i.e., 'Pd') stands for?
8) A paragraph discussing future extensions and applications of this work can be added to the conclusion.
9) Most of the references included in this work are a bit dated, thus, more recent references can be added. The authors may want to consider the following references among others:
[1] Fotouhi, A., Ding, M., & Hassan, M. (2021). Deep Q-Learning for Two-Hop Communications of Drone Base Stations. Sensors, 21(6), 1960.
[2] Pliatsios, D., Sarigiannidis, P., Goudos, S. K., & Psannis, K. (2020). 3D Placement of Drone-Mounted Remote Radio Head for Minimum Transmission Power under Connectivity Constraints. IEEE Access, 8, 200338-200350.
[3] Li, Z., Wang, Y., Liu, M., Sun, R., Chen, Y., Yuan, J., & Li, J. (2019). Energy efficient resource allocation for UAV-assisted space-air-ground Internet of remote things networks. IEEE Access, 7, 145348-145362.
[4] Li, B., Fei, Z., Zhang, Y., & Guizani, M. (2019). Secure UAV communication networks over 5G. IEEE Wireless Communications, 26(5), 114-120.
[5] Yang, H., & Xie, X. (2019). Energy-efficient joint scheduling and resource management for UAV-enabled multicell networks. IEEE Systems Journal, 14(1), 363-374.
[6] Chen, Y., Zhao, N., Ding, Z., & Alouini, M. S. (2018). Multiple UAVs as relays: Multi-hop single link versus multiple dual-hop links. IEEE Transactions on Wireless Communications, 17(9), 6348-6359.
[7] Illi, E., El Bouanani, F., Sofotasios, P. C., Muhaidat, S., da Costa, D. B., Ayoub, F., & Al-Fuqaha, A. (2020). Analysis of asymmetric dual-hop energy harvesting-based wireless communication systems in mixed fading environments. IEEE Transactions on Green Communications and Networking, 5(1), 261-277.
[8] Mogadala, V. K., Gottapu, S. R., & Chapa, B. P. (2019, March). Dual Hop Hybrid FSO/RF based Backhaul Communication System for 5G Networks. In 2019 International Conference on Wireless Communications Signal Processing and Networking (WiSPNET) (pp. 229-232). IEEE.
[9] Rahman, M. A., Lee, Y., & Koo, I. (2018). Energy-efficient power allocation and relay selection schemes for relay-assisted d2d communications in 5g wireless networks. Sensors, 18(9), 2865.
[10] Selmi, S., & Bouallègue, R. (2020, September). Joint Spectral and Energy Efficient Multi-hop D2D Communication Underlay 5G Networks. In 2020 International Conference on Software, Telecommunications and Computer Networks (SoftCOM) (pp. 1-6). IEEE.

Reviewer 2 Report

This study explored the likelihood ratio test (LRT) with cascade channel frequency response which is optimal according to the Neyman-Pearson theorem. Simulation results demonstrate that the proposed power allocation strategy outperforms the equal power allocation in terms of FAR and MDR. The work is well organized and appropriately carried out.  I have looked at the mathematics and it looks sound. At this p point, I recommend the following comments to be revised and be expressed more analytically in a revised version, as this current work is definitely worth publication.  

1) In the Introduction, the authors should summarize their contribution in the introduction section using some bullets or numbers briefly.
2)  It would be better to compare the performance of the proposed design with other published work in Table in a new section called Section 2 "Related works".
3) References should be provided for the equations which were borrowed from the literature.
4)In Table 1, Why you are considered Carrier frequency 2 GHz, why not 2.4 GHz for LTE technology? Also, Subcarrier interval are very tight why not 1.25, 5, 10, 15 or 20 MHz?
5)The parameters given in Table 1 need references to support them.
6)I recommend providing the figures in the result section with colors to distinguish the lines.
7)What are the limitations of this study? and How could/should futures studies improve the model?

Reviewer 3 Report

This work presents a channel-based physical layer authentication system for dual-hop wireless networks. Optimal power with respect to the sum of FAR and MDR is also derived. Numerical simulations show the validity of the proposed solution. 

The results are plotted vs SNR, it would be good to see how the results vary with the thresholds.

Equation (40) and (42) are the same. 

Minor corrections:

Page 2 - "Power P1 , realay recieves and forwards to Bob" -> "Power P1 , relay receives and forwards to Bob"

Page 5 - "and then copared with threshold" -> "and then compared with threshold"

Page 6 - "propabality" -> "probability" 

Page 7 - "Based on formula formula (21 )" -> "Based on formula (21 )"

Page 8 - "HRB( k,  t 1) ≈ HRB( k,  t 1)," -> "HRB( k,  t) ≈ HRB( k,  t 1),"

Round 2

Reviewer 2 Report

The authors have addressed the comments.

Author Response

Thank you for your comments.